# Correlation between the Biodistribution of Bovine Leukemia Virus in the Organs and the Proviral Load in the Peripheral Blood during Early Stages of Experimentally Infected Cattle

**DOI:** 10.3390/pathogens12010130

**Published:** 2023-01-12

**Authors:** Junko Kohara, Lanlan Bai, Shin-nosuke Takeshima, Yuki Matsumoto, Tsunao Hirai, Yoko Aida

**Affiliations:** 1Animal Health Group, Animal Research Center, Hokkaido Research Organization, Shintoku 081-0038, Japan; 2Virus Infectious Diseases Unit, RIKEN, 2-1 Hirosawa, Wako 351-0198, Japan; 3Graduate School of Science and Engineering, Iwate University, Morioka 020-8551, Japan; 4Department of Food and Nutrition, Jumonji University, 2-1-28 Sugasawa, Niiza 352-8510, Japan; 5Laboratory of Global Infectious Diseases Control Science, Graduate School of Agricultural and Life Sciences, The University of Tokyo, 1-1-1 Yayoi, Bunkyo-ku, Tokyo 113-8657, Japan

**Keywords:** bovine leukemia virus, biodistribution, organ, proviral load, peripheral blood, BLV-CoCoMo-qPCR-2, experimentally infected cattle

## Abstract

Bovine leukemia virus (BLV) is the etiological agent of enzootic bovine leukosis. However, the propagation and distribution of BLV after primary infection still need to be fully elucidated. Here, we experimentally infected seven cattle with BLV and analyzed the BLV proviral load (PVL) in the blood and various organs. BLV was first detected in the blood of the cattle after one week, and the blood PVL increased for three weeks after infection. The PVL was maintained at a high level in five cattle, while it decreased to a low or medium level in two cattle. BLV was distributed in various organs, such as the heart, lung, liver, kidney, abomasum, and thymus, and, notably, in the spleen and lymph nodes. In cattle with a high blood PVL, BLV was detected in organs other than the spleen and lymph nodes, whereas in those with a low blood PVL, BLV was only detected in the spleen and lymph nodes. The amount of BLV in the organs was comparable to that in the blood. Our findings point to the possibility of estimating the distribution of BLV provirus in organs, lymph nodes, and body fluids by measuring the blood PVL, as it was positively correlated with the biodistribution of BLV provirus in the body of BLV infection during early stages.

## 1. Introduction

The bovine leukemia virus (BLV) is classified under the genus *Deltaretrovirus* in the family *Retroviridae* and is the etiological agent of enzootic bovine leukosis (EBL). Most BLV-infected cattle remain clinically silent during the aleukemic stage. Following the aleukemic stage, which ranges from months to years, approximately 30% of BLV-infected cattle develop polyclonal non-neoplastic B-cell lymphocytosis (persistent lymphocytosis), and only 1–5% of BLV-infected cattle develop malignant B-cell lymphomas [1].

The BLV genome integrates into the host genome as provirus [2,3], even in the absence of detectable BLV antibodies, and it can be amplified after a period of latency [2,3,4]. Therefore, diagnostic BLV polymerase chain reaction (PCR) techniques that detect the integrated BLV proviral genome within the host genome are used in addition to serological techniques for the diagnosis of BLV infections [5]. Indeed, previous studies have shown that the proviral load (PVL) is an important index for estimating the stage of BLV infection because it is associated with disease progression [5,6,7,8], BLV infectivity as assessed via syncytium formation [9,10,11], the lymphocyte count [12], viral biokinetics [13], and virus shedding into the salivary and nasal secretions [14], and milk [15,16]. Indeed, previous reports have shown that the BLV provirus can be detected in the milk, nasal mucus, and saliva of dairy cattle with PVLs of >10,000, 14,000, and 18,000 copies/10^5^ cells in their blood samples, respectively [14,15]. These results suggest that a PVL of approximately 10,000 copies/10^5^ cells in the blood might be an indicator of efficient BLV spreading throughout the whole body, which is a relatively high number. Therefore, a BLV PVL of 10,000 copies/10^5^ cells has been set as a threshold to distinguish between high- and low-PVL cows [17,18]. Thus, the PVL is considered a major diagnostic index for estimating the BLV transmission risk [19].

Natural or iatrogenic transmission of BLV primarily involves the transfer of infected cells via the blood. In experimental infections in sheep, which are highly susceptible to BLV, and in which tumors are formed at a high frequency, BLV-infected cells became detectable in the blood during the second week after virus infection and decreased rapidly thereafter [20,21]. Although the major target of the virus is B-lymphocytes, BLV can persist in the cells of the monocyte/macrophage lineage [22,23]. However, in terms of experimental infections in cattle, the BLV provirus was first detected during the first and second weeks after virus infection, it peaked at 30 d post-inoculation in all animals, and steadily decreased after seroconversion [24]. However, the propagation and distribution of BLV after primary infection remain to be fully elucidated.

In experimentally infected calves, which were at an early stage of BLV infection, BLV proviral DNA was detected in predilection sites of tumors, such as the spleen, uterus, liver, kidney, abomasum, and lymph nodes [25]. In EBL cattle, the BLV PVL in the blood, spleen, and lymph nodes was higher than that in BLV-infected cattle without clinical signs [7]. These findings suggest that the distribution and quantity of the BLV provirus could be valuable information for predicting EBL.

Consequently, in this study, to reveal the correlation between the biodistribution of BLV in the organs and PVL in cattle, we experimentally infected BLV-infected cells into cattle and analyzed the PVL of BLV in the blood and various organs of the cattle using the quantitative real-time PCR (qPCR) assay BLV-CoCoMo-qPCR-2 [6,26].

## 2. Materials and Methods

### 2.1. Experimental Animals

Five Japanese Black cattle (4–5 months old) and two Holstein steers (12–13 months old) were housed in individual stalls at the Animal Research Center of Hokkaido (Table 1). The experimental cattle were not infected with BLV at the start of the study, as determined by a BLV-specific PCR assay [27]. The experimental cattle were infected with BLV-infected white blood cells that were prepared from a donor cow persistently infected with BLV. Each cattle received approximately 8 × 10^7^ cells containing 4 × 10^7^ copies of the BLV provirus, as determined by the BLV-CoCoMo-qPCR-2 assay.

### 2.2. Collection of Blood and Organs

Blood samples were collected for up to 30 and 40 weeks after BLV infection. The number of peripheral blood lymphocytes was measured using an automated blood cell counter (Cell Dyn 3700; Abbott Laboratories, Chicago, IL, USA). Between seven and nine months after infection, the experimental cattle were euthanized after the last blood sampling, and necropsy was performed to obtain organ samples. The following organs were collected: the right auricle of the heart, liver, spleen, kidney, lung, abomasum, thymus, and mandibular, and superficial cervical, subiliac, and mesenteric lymph nodes. Blood cells were removed from the organ samples.

### 2.3. DNA Extraction from the Blood and Organs

DNA was extracted from EDTA-treated whole blood samples using a Wizard Genomic DNA Purification Kit (Promega Corporation, Tokyo, Japan). The organ samples were minced with sterile scissors, and the DNA was extracted using the DNeasy Blood and Tissue Kit (QIAGEN, Tokyo, Japan) according to the manufacturer’s instructions.

### 2.4. Detection and Quantification of the Bovine Leukemia Virus Proviral Load

BLV PVLs were quantified using BLV-CoCoMo-qPCR-2 with THUNDERBIRD Probe qPCR Mix (Toyobo, Tokyo, Japan), as previously described [6,26,28,29]. The long terminal repeat (LTR) region of the BLV was amplified using the degenerate primer pair CoCoMo-FRW and CoCoMo-REV. Additionally, FAM-LTR was used as the probe. The *BoLA-DRA* gene (internal control) was amplified using the primer pair DRA-F and DRA-R, with FAM-DRA being used as the probe. Finally, the PVL was calculated using the following formula: (number of BLV LTR copies/number of *BoLA-DRA* copies) × 10^5^ cells.

### 2.5. Detection of the Anti-Bovine Leukemia Virus Antibodies in the Serum Samples

Anti-BLV antibodies were detected using ELISA kits, according to the manufacturer’s instructions (JNC Inc., Tokyo, Japan). 

### 2.6. Statistical Analysis

Group comparisons were performed using one-way analysis of variance followed by Tukey’s multiple comparison tests using GraphPad Prism version 5. Statistical significance was set at *p* < 0.05. The correlation coefficient (R) between the BLV PVL in organs and blood was calculated using Excel with the Pearson function.

## 3. Results

### 3.1. Bovine Leukemia Virus Proviral Load in the Blood of Bovine Leukemia Virus-Infected Cattle

To estimate the distribution of BLV in the organs of experimentally infected cattle with different BLV PVLs in their peripheral blood, seven cattle, including five Japanese Black cattle and two Holstein cattle, were experimentally infected with the same amount of BLV from a single infected cow (Table 1). All cattle were positive for anti-BLV antibodies, as determined by ELISA. The clinical stage of BLV infection in these cattle was evaluated according to the lymphocyte count (per μL) and age of the animal [30]. All cattle were categorized as BLV-infected, but clinically and hematologically were normal cattle. 

As shown in Figure 1, BLV proviral DNA was first detected in the blood of the experimental cattle one or two weeks after infection with BLV-infected white blood cells. The BLV PVL increased to more than 3.0 × 10^4^ copies/10^5^ cells three weeks after infection with BLV, and the BLV PVL varied among the experimentally infected cattle. In animals #A, #B, #C, #D, and #E, a high PVL (>10,000 copies/10^5^ cells) was maintained for up to 30–40 weeks of infection (Figure 1 and Table 1). In contrast, the PVL in animal #G reached 5.8 × 10^4^ copies/10^5^ cells 3 weeks after infection, then decreased to less than 1.0 × 10^2^ copies/10^5^ cells 30 weeks after infection. In animal #F, the PVL decreased after 12 weeks of infection and remained between 2.0 × 10^3^ copies and 5.0 × 10^3^ copies/10^5^ cells. Thus, animals #A, #B, #C, #D, and #E had a high PVL (more than 2.2 × 10^4^ copies/10^5^ cells), animal #F had a medium PVL (2.3 × 10^3^ copies/10^5^ cells), and animal #G had a low PVL (5.3 × 10 copies/10^5^ cells; Table 1).

### 3.2. Distribution of Bovine Leukemia Virus Proviral DNA in the Organs

Seven to nine months after infection, the seven experimentally infected cattle were euthanized and used for the detection of proviral DNA in the organs. Grossly, no macroscopic lesions were found at necropsy in any of the seven experimentally infected cattle. 

As shown in Table 2, in animals #A, #B, #C, #D, and #E, which had a high blood PVL, BLV proviral DNA was detected by BLV-CoCoMo-qPCR-2 in all examined organs, including the heart, lung, liver, kidney, abomasum, thymus, and spleen. Notably, the blood PVL was high in the spleen and lymph nodes, with PVLs ranging from 5.3 × 10^2^ copies/10^5^ cells to 2.4 × 10^4^ copies/10^5^ cells, while the PVL was low in the thymus, with PVLs ranging from 19 copies/10^5^ cells to 68 copies/10^5^ cells. In animal #G, which had a low blood PVL, BLV proviral DNA was only detected in the spleen and lymph nodes but was not detected in the other organs. In cattle #F, which had a medium blood PVL, BLV proviral DNA was not detected in the thymus but in all other organs.

In the spleen of the experimental cattle, the PVL ranged from 4.3 × 10 copies/10^5^ cells to 2.4 × 10^4^ copies/10^5^ cells, and the average PVL was 1.0 × 10^4^ copies/10^5^ cells (Table 2). In the lymph nodes of the experimentally infected cattle, the PVL ranged from 2.0 × 10 copies/10^5^ cells to 1.4 × 10^4^ copies/10^5^ cells, and the average PVL ranged from 2.5 × 10^3^ copies/10^5^ cells to 4.5 × 10^3^ copies/10^5^ cells. In the heart, lung, liver, kidney, and abomasum of the experimentally infected cattle, the PVL ranged from <10 copies/10^5^ cells to 9.0 × 10^3^ copies/10^5^ cells, and the average PVL ranged from 3.8 × 10^2^ copies/10^5^ cells to 4.2 × 10^3^ copies/10^5^ cells. Lastly, in the thymus, the PVL ranged from <10 copies/105 cells to 6.8 × 10 copies/10^5^ cells, and the average PVL was 3.0 × 10 copies/10^5^ cells, indicating that the PVL of the thymus was the lowest among the organs that were analyzed in this study. In addition, in terms of animals #B–F, the PVLs in the spleen were higher than those in the mandibular, superficial cervical, subiliac, and mesenteric lymph nodes. However, animals #A and #D had similar PVLs in the mesenteric lymph nodes to those in the spleen, which differed from those in the superficial cervical, subiliac, and mesenteric lymph nodes. Thus, in all cattle, PVLs in the spleen and lymph nodes tended to be higher than those in the other organs.

Furthermore, to analyze the correlation between blood PVL and PVL in all examined organs, including the heart, lung, liver, kidney, abomasum, thymus, spleen, and lymph nodes, we constructed a scatter graph and performed linear regression analysis using the PVL data in the blood and each organ of the animals (Figure 2). Interestingly, a strong positive correlation was observed between the PVL in blood and that in all examined organs; the Spearman’s rank correlation coefficient (R) ranged from 0.6787 to 0.9639 (R = 0.8129 ± 0.1031). These results indicate a correlation between the blood PVL and the biodistribution of proviral DNA.

### 3.3. Comparison of the Distribution of Bovine Leukemia Virus Proviral DNA in the Organs and the Proviral Load in the Blood

The PVLs in the spleen and lymph nodes were compared to those in the blood at necropsy (Figure 3). In animals #A–F, which had a high or medium blood PVL, the blood PVLs were significantly higher than those in the spleen and lymph nodes. In contrast, in animal #G, which had a particularly low blood PVL, the PVL level in the spleen and lymph nodes was the same as that in the blood.

## 4. Discussion

In this study, we first demonstrated that BLV proviral DNA was distributed in various organs, such as the heart, lung, liver, kidney, abomasum, and thymus, and, notably, it was highly distributed in the spleen and lymph nodes. This result is supported by a previous report, where BLV proviral DNA was detected in the spleen, uterus, liver, kidney, abomasum, and lymph nodes [26]. Second, we showed that there was a correlation between the blood PVL and the biodistribution of proviral DNA. Specifically, in cattle with a high blood PVL, proviral DNA was detected in the spleen, lymph nodes, and other organs, and the level of the blood PVL was higher than that in the organs. In cattle with a low blood PVL, proviral DNA was only detected in the spleen and lymph nodes, and the blood PVL level was comparable to that of these organs. Third, the spleen has previously been reported to be a secondary lymphoid organ other than the lymph nodes and to remove old erythrocytes from the blood circulation and blood-borne microorganisms or cellular debris [31]. However, in our study, we detected BLV proviral DNA in the spleen of all animals, and the PVL in the spleen was generally higher than that in the other organs. This finding suggests that the virus was trapped and propagated in the spleen. This finding correlates well with results reported previously that the initial site of BLV replication was in the spleen rather than the regional lymph node nearest to the inoculation site [32]. Therefore, lymph nodes are important sites for the establishment of infection by lymphotropic human and simian immunodeficiency viruses, and they are virus reservoirs during the asymptomatic stages of infection [33,34]. 

The change in the BLV PVL in the blood of the infected cattle varied depending on the cattle, reaching a peak 3 weeks after BLV infection. Between 30 and 40 weeks after infection, the infected cattle could be separated as follows: a group with a high PVL (>10,000 copies of provirus/10^5^ cells; animals #A, #B, #C, #D, and #E), one animal with a medium PVL (between 100 and 10,000 copies of provirus/10^5^ cells; animal #F), and one animal with a low PVL (<100 copies of provirus/10^5^ cells; animal #G). In animal #G, the PVL increased to a high level during early infection, then decreased to a very low level or the BLV level could not be detected in blood. In the group with a high blood PVL, BLV proviral DNA was detected in all organs, including the heart, lung, liver, kidney, abomasum, thymus, and spleen. In contrast, in the animal with a low blood PVL, proviral DNA was only detected in the spleen and lymph nodes. Therefore, this study clearly shows that the extent of the distribution of BLV proviral DNA in the organs was associated with the PVL levels in the blood of the BLV-infected cattle. In addition, it was previously shown that the BLV provirus can be detected in body fluids, such as milk, nasal mucus, and saliva samples of dairy cattle, with PVLs of more than 10,000, 14,000, and 18,000 copies/10^5^ cells in blood samples, respectively [14,15]. Thus, our present and previous results clearly show that the PVLs in peripheral blood may be a marker for the extent of the distribution of BLV proviral DNA in organs, lymph nodes, and body fluids, such as milk, and nasal and salivary secretions.

There were also individual differences in the BLV-infected cattle, such as variations in the distribution of BLV in the organs and PVL in the peripheral blood. This may be explained by the fact that the BLV PVL has been associated with the *BoLA* genotype in Holstein-Friesian cattle and Japanese Black cattle [17,18,35,36,37,38]. Interestingly, out of four Japanese Black cattle with a high PVL, three cattle, animals #A, #B, and #C, carried the BLV-susceptible-associated *BoLA-DRB3* allele (*BoLA-DRB3*016:01*), which has been reported in Japanese Black cattle [34], supporting that *BoLA-DRB3*016:01* was associated with a high PVL. The remaining Japanese Black cattle with a high PVL (#D) carried the *BoLA-DRB3*005:03* allele, which is positively associated with BLV-induced lymphoma [36]. On the other hand, one Holstein cattle with a high PVL had the *BoLA-DRB3* allele (*BoLA-DRB3*012:01*), which is positively associated with lymphoma development in Holstein cattle. In contrast, the Japanese Black cattle with a low PVL (#G) carried the *BoLA-DQA1*002:04* allele, which is associated with low PVLs [34], while the remaining Holstein cattle (#F) did not harbor known *BoLA-DRB3* alleles associated with the BLV PVL. However, although Holstein cattle #F carried the *BoLA-DRB3*015:01* allele, which is associated with a high PVL in Holstein cattle [16], this individual only had a medium PVL, with 2.3 × 10^2^ copies/10^5^ cells in the blood sample. This result was strongly supported by the fact that the alleles of *BoLA-DQA*1, which is another *BoLA* class II locus, were first identified to determine the BLV PVL in Japanese Black cattle [36]. Further support is provided by research showing that single nucleotide polymorphisms in the bovine MHC region are associated with the BLV PVL based on a whole genome association study in Japanese Black cattle [39]. These results suggest that understanding the host factors that are associated with the BLV PVL may provide important clues for controlling the spread of BLV in cattle.

Overall, our study indicated that if BLV had not been eliminated from the blood after primary infection, the virus was distributed in the organs throughout the body. Even if the virus was strongly eliminated from the blood by the immune response, the virus remained in particular organs, such as the spleen and lymph nodes. In addition, our results showed the possibility of estimating the distribution of BLV proviral DNA in organs, lymph nodes, and body fluids by measuring the BLV PVL in blood, as the blood PVL was positively correlated with the biodistribution of the BLV provirus in the body of BLV infection during early stages.

## 5. Conclusions

In this study, seven BLV-free calves were intravenously inoculated with white blood cells that were prepared from a BLV-infected donor. BLVs were distributed in various organs, such as the heart, lung, liver, kidney, abomasum, and thymus, and, notably, they were highly distributed in the spleen and lymph nodes. In the five cattle, in which the PVL was maintained at a high level, BLV was detected in organs other than the spleen and lymph nodes. In contrast, in cattle with a low blood PVL, the BLV provirus was only detected in specific organs, including the spleen and lymph nodes, and the level of the virus in the organs was comparable to that in the blood. Thus, the results indicate that BLV develops differently within infected cattle, resulting in cattle with either a high load or low load of the virus in the blood. Moreover, BLV resides in the spleen and lymph nodes, even if eliminated from the blood. Our results also demonstrated that the blood PVL was positively correlated with the biodistribution of the BLV provirus in the body of BLV infection during early stages, indicating that the distribution of BLV proviral DNA could possibly be estimated using the blood BVL PVL.

## Figures and Tables

**Figure 1 pathogens-12-00130-f001:**
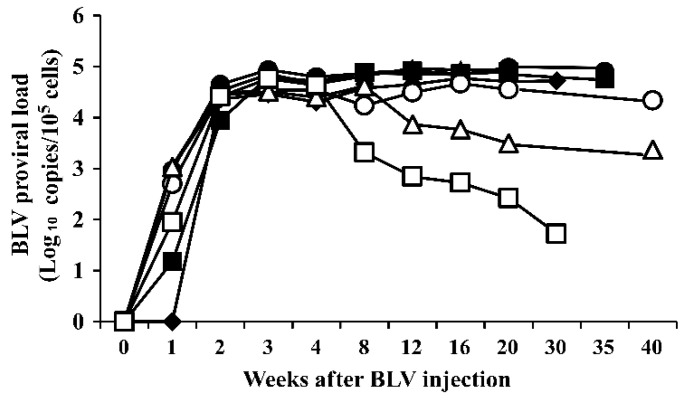
Changes in the BLV proviral load in blood of the experimentally infected cattle. ●: animal #A ▲: animal #B, ■: animal #C, ◆: animal #D, ○: animal #E, △: animal #F, □: animal #G.

**Figure 2 pathogens-12-00130-f002:**
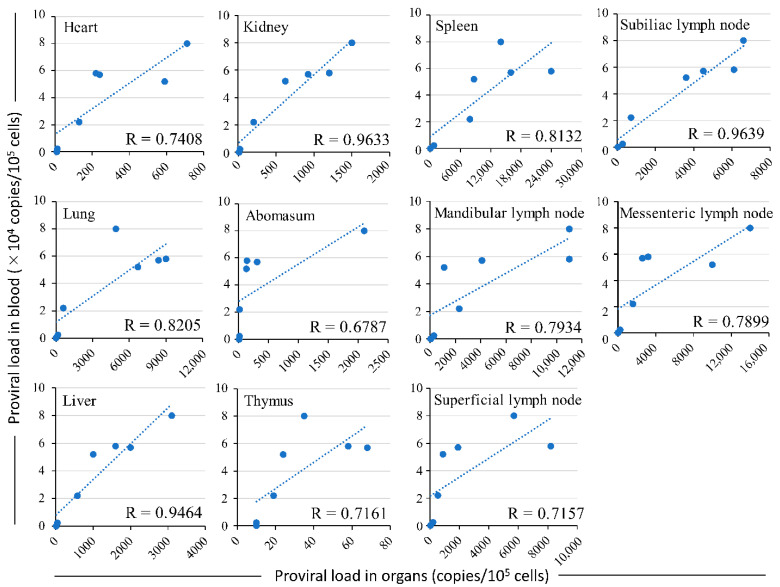
Correlation between the blood PVL and that in all examined organs, including the heart, lung, liver, kidney, abomasum, thymus, spleen, and lymph nodes. The BLV PVL (expressed as the number of copies of proviral load per 10^5^ cells) in organs and peripheral blood of the experimentally infected cattle was evaluated using CoCoMo-qPCR-2.

**Figure 3 pathogens-12-00130-f003:**
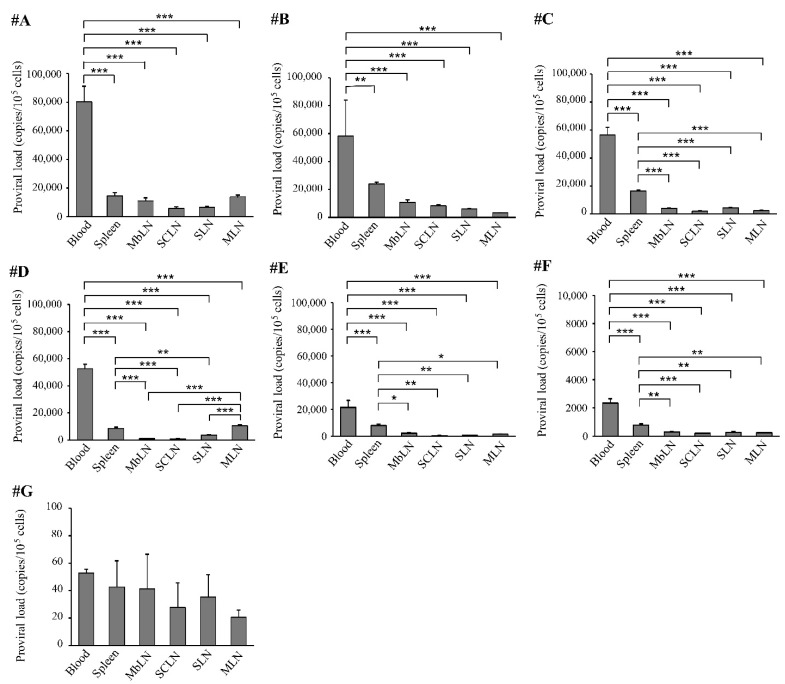
BLV proviral load in the blood, spleen, and lymph nodes. Data are expressed as means from three independent experiments (* *p* < 0.05, ** *p* < 0.01, and *** *p* < 0.001). MbLN; mandibular lymph node, SCLN; superficial cervical lymph node, SLN; subiliac lymph node, MLN; and mesenteric lymph node.

**Table 1 pathogens-12-00130-t001:** Breed, BLV proviral load in blood, and number of peripheral blood lymphocytes of experimentally infected cattle.

Cattle ^a^	Breed	Age ^b^ (Month) Start End	Sex	BLV Proviral Load in Blood (Copies/10^5^ Cells) ^c^	Number of PBL (/μL) ^d^	Antibodies to BLV ^e^
(1) High proviral load cattle						
#A	Japanese Black	4–12	♂	8.0 × 10^4^	9.2 × 10^3^	+
#B	Japanese Black	5–14	♂	5.8 × 10^4^	8.8 × 10^3^	+
#C	Japanese Black	5–13	♂	5.7 × 10^4^	8.7 × 10^3^	+
#D	Japanese Black	5–12	♀	5.2 × 10^4^	8.6 × 10^3^	+
#E	Holstein-Friesian	12–22	♂	2.2 × 10^4^	7.1 × 10^3^	+
(2) Medium proviral load cattle						
#F	Holstein-Friesian	13–23	♂	2.3 × 10^3^	5.2 × 10^3^	+
(3) Low proviral load cattle						
#G	Japanese Black	5–12	♀	5.3 × 10	4.9 × 10^3^	+

^a^ Five BLV-negative Japanese Black calves and two Holstein steers were experimentally infected intravenously with blood containing a proviral load of 4 × 10^7^ proviral copies, as determined by CoCoMo-qPCR-2 assay. ^b^ Age, start; age at the time of the BLV infection, End; age at the time of necropsy. ^c^ Proviral load (expressed as the number of copies of provirus per 10^5^ peripheral blood mononuclear cells) at the time of necropsy was evaluated using CoCoMo-qPCR-2. ^d^ Number of peripheral blood lymphocytes (PBLs) of the cattle at the time of necropsy. ^e^ ELISA (enzyme-linked immunosorbent assay) to detect anti-BLV antibodies 4 weeks after BLV infection using the BLV ELISA kit, + = BLV positive serum.

**Table 2 pathogens-12-00130-t002:** BLV proviral load in organs and peripheral blood of the experimentally infected cattle.

Organs and Peripheral Blood	Experimental Infected Cattle							
	#A	#B	#C	#D	#E	#F	#G	Average
Right auricle of the heart	7.1 × 10^2^	2.2 × 10^2^	2.4 × 10^2^	5.9 × 10^2^	1.3 × 10^2^	1.3 × 10	<10	2.7 × 10^2^
Lung	4.9 × 10^3^	9.0 × 10^3^	8.4 × 10^3^	6.7 × 10^3^	6.0 × 10^2^	1.6 × 10^2^	<10	4.2 × 10^3^
Liver	3.1 × 10^3^	1.6 × 10^3^	2.0 × 10^3^	1.0 × 10^3^	5.8 × 10^2^	4.44 × 10	<10	1.2 × 10^3^
Kidney	1.5 × 10^3^	1.2 × 10^3^	9.2 × 10^2^	6.2 × 10^2^	2.0 × 10^2^	2.0 × 10	<10	6.3 × 10^2^
Abomasum	2.1 × 10^3^	1.4 × 10^2^	3.1 × 10^2^	1.3 × 10^2^	1.7 × 10	1.4 × 10	<10	3.8 × 10^2^
Thymus	3.5 × 10	5.8 × 10	6.8 × 10	2.4 × 10	1.9 × 10	<10	<10	3.0 × 10
Spleen	1.4 × 10^4^	2.4 × 10^4^	1.6 × 10^4^	8.7 × 10^3^	7.9 × 10^3^	7.7 × 10^2^	4.3 × 10	1.0 × 10^4^
Mandibular lymph node	1.1 × 10^4^	1.1 × 10^4^	4.1 × 10^3^	1.1 × 10^3^	2.3 × 10^3^	3.0 × 10^2^	4.1 × 10	4.2 × 10^3^
Superficial cervical lymph node	5.7 × 10^3^	8.2 × 10^3^	1.9 × 10^3^	8.7 × 10^2^	5.3 × 10^2^	2.0 × 10^2^	2.8 × 10	2.5 × 10^3^
Subiliac lymph node	6.6 × 10^3^	6.1 × 10^3^	4.5 × 10^3^	3.6 × 10^3^	7.2 × 10^2^	2.8 × 10^2^	3.6 × 10	3.1 × 10^3^
Mesenteric lymph node	1.4 × 10^4^	3.2 × 10^3^	2.6 × 10^3^	1.0 × 10^4^	1.6 × 10^3^	2.4 × 10^2^	2.0 × 10	4.5 × 10^3^
Peripheral Blood	8.0 × 10^4^	5.8 × 10^4^	5.7 × 10^4^	5.2 × 10^4^	2.2 × 10^4^	2.3 × 10^3^	5.3 × 10	3.9 × 10^4^

BLV proviral load (expressed as the number of copies of provirus per 10^5^ cells) was evaluated using CoCoMo-qPCR-2.

## Data Availability

All relevant data are included in the paper.

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
