# Peer review of "Correlation between the Biodistribution of Bovine Leukemia Virus in the Organs and the Proviral Load in the Peripheral Blood during Early Stages of Experimentally Infected Cattle"

_pathogens, 2023, doi:10.3390/pathogens12010130_

Round 1

Reviewer 1 Report

The authors examined how bovine leukemia virus (BLV) was distributed in cattle experimentally inoculated with BLV-infected blood. They concluded that BLV proviral load in peripheral blood was indicative of the BLV distribution in the body. The experiments are simple, but some valuable results are included.

Major and minor comments,

Line 22, "injection" -> challenge? infection? intravenous inoculation? Similarly, in lines 59, 61, 64, 65, 75, 84, and maybe more.

Line 22: Clarify if "PVL" means PVL in peripheral blood or organs or both.

Line 27: "the same as" -> similar? comparable?

Line 41: Revise the sentence, "BLV provirus remains is integrated into…".

Line 47: [5-7] -> Kobayashi T. et al., 2019. Prev. Vet. Med. should be included.

Line 47: The parentheses are not complete. Only “(“ appears.

Line 71: “, which were clinically healthy -> without clinical signs

Line 77: Delete “which uses …and is termed”.

Line 85: Delete “, which was”.

Line 90-91: The sentence is unclear.

Line 94: after blood sampling -> after the last blood sampling?

Line 95: “, which was followed by a necropsy and organ sampling.” -> “and a necropsy was performed to obtain organ samples.

Line 106-107: The sentence is overlapped with line 100-101.

Line 106-108: Revise the sentence.

Line 108-112. Provide the equation for PVL

Line 121: P<0.0?

Line 127: Delete “, which was taken”

Line 128-129: Delete “an enzyme-linked immunosorbent assay”. ELISA appeared earlier already.

Line 130: “age of the animal” -> EC-leukosis key?

Line 135: cattle # -> animal #. “cattle” is generally used as a group.

Line 154: The closed circle is too small. It should be the same size as the open circle.

Line 154: Fig 1. Experimental cattle -> experimentally infected cattle. Avoid “experimental cattle”. It appears in other places (line136 and so on).

Line 162: ”all the organs, such as” -> “all examined organs including”?

Line 166: “However”, “particularly”-> these wordings are not appropriate.

Line 204: "This result is supported by" others. This could mean this study was a simple confirmation of previous studies. Novelty in this study should be therefore stated. Also, "Sites of in vivo replication of bovine leukemia virus in experimentally infected cattle" by M. J. Van Der Maaten, J. M. Miller 1978.  should be cited.

Line 206: The "correlation" should be analyzed using such as Pearson's Coefficient.

Line 209-211: The sentence is confusing and needs revision.

Line 214-216: The sentence is confusing and needs revision.

Line 227: “For example”?

Line 227, 229: Delete “peripheral”

Line 230: There is a double space between blood and PVL.

Line 232: “Similarly”?

Line 233: “matched blood samples”?

Line 232-235: Organize the sentence according to what should be emphasized. It could be divided into two sentences.

Line 243: BLV-sensitive -> BLV-susceptible

Line 247: “In addition”?

Line 248: “BoLA-DRB3 allele (BoLA-DRB3*1201) -> BoLA-DRB3*012:01

Line 239-261: This paragraph contains information on DRB3 alleles for cattle used in this study. The information should be included in Table 1 and the method for typing should be described in Materials and Methods. Also, consider changing the names of alleles according to the latest nomenclature site. For example, DRB3*1601 -> DRB3*016:01.

Line 250: is related -> is associated

Line 264: “strongly eliminated”?

Line 276: “specific organs, including the spleen and lymph nodes”. What are the specific organs?

Line 277. “the level of the virus in the organs was the same as that in the blood”. Was it the same? comparable? similar?

Author Response

Answer to the comments of reviewer #1:

Thank you very much for the helpful and constructive comments. We have amended the manuscript with these comments and our point-by-point responses are set out below.

The authors examined how bovine leukemia virus (BLV) was distributed in cattle experimentally inoculated with BLV-infected blood. They concluded that BLV proviral load in peripheral blood was indicative of the BLV distribution in the body. The experiments are simple, but some valuable results are included.

Major and minor comments,

Line 22, "injection" -> challenge? infection? intravenous inoculation? Similarly, in lines 59, 61, 64, 65, 75, 84, and maybe more.

Answer: In response to the reviewer's comment, we have changed from "injection" to "infection" in lines 22, 59, 61, 64, 65, 75, 83, 132, 133, 137, 147, 151, 239, and 240 in the revised manuscript.

Line 22: Clarify if "PVL" means PVL in peripheral blood or organs or both.

Answer: PVL in line 22 means PVL in blood. Therefore, in response to the reviewer's comment, we have changed from "PVL" to "blood PVL" in Line 22 in the revised manuscript.

Line 27: "the same as" -> similar? comparable?

Answer: In response to the reviewer's comment, we have changed from "same" to "comparable" in line 27 of the revised manuscript.

Line 41: Revise the sentence, "BLV provirus remains is integrated into…".

Answer: In response to the reviewer's comment, we have changed the sentence to "The BLV genome integrates into the host genome as provirus" in line 41 in the revised manuscript.

Line 47: [5-7] -> Kobayashi T. et al., 2019. Prev. Vet. Med. should be included.

Answer: In response to the reviewer's comment, we have cited Kobayashi T. et al., 2019. Prev. Vet. Med in References as reference number 8 and changed the order of references in the revised manuscript.

Line 47: The parentheses are not complete. Only "(" appears.

Answer: We apologize for our incomplete description making the reviewer confusing. In response to the reviewer's comment, we have changed the sentence to "it is associated with disease progression [5-8], BLV infectivity as assessed via syncytium formation [9-11], the lymphocyte count [12], viral biokinetics [13], and virus shedding into the salivary and nasal secretions [14] and milk [15,16]." in lines 47-49 in the revised manuscript.

Line 71: ", which were clinically healthy -> without clinical signs

Answer: In response to the reviewer's comment, we have changed from which were clinically healthy to "the BLV-infected cattle without clinical signs" in lines 71 of the revised manuscript.

Line 77: Delete "which uses …and is termed".

Answer: In response to the reviewer's comment, we have deleted this sentence in line 77 in the original manuscript.

Line 85: Delete ", which was".

Answer: In response to the reviewer's comment, we have deleted these words in line 85 of the original manuscript.

Line 90-91: The sentence is unclear.

Answer: In response to the reviewer's comment, we have changed the sentence to "Blood samples were collected for up to 30 and 40 weeks after BLV infection." in line 89 in the revised manuscript.

Line 94: after blood sampling -> after the last blood sampling?

Answer: In response to the reviewer's comment, we have changed from "after the last blood sampling," in lines 92-93 in the revised manuscript.

Line 95: ", which was followed by a necropsy and organ sampling." -> "and a necropsy was performed to obtain organ samples.

Answer: In response to the reviewer's comment, we have changed from "which was followed by a necropsy and organ sampling" to ", and a necropsy was performed to obtain organ samples." in lines 93 in the revised manuscript.

Line 106-107: The sentence is overlapped with lines 100-101.

Answer: We apologize for our overlapped description. In response to the reviewer's comment, we have deleted the overlapped sentence in lines 106-107 in the original manuscript.

Line 106-108: Revise the sentence.

Answer: In response to the reviewer's comment, we have revised it to "BLV PVLs were quantified using BLV-CoCoMo-qPCR-2 with THUNDERBIRD Probe qPCR Mix (Toyobo, Tokyo, Japan), as previously described [6,25,27,28]." In lines 105-106 in the revised manuscript.

Line 108-112. Provide the equation for PVL

Answer: In response to the reviewer's comment, we have added an explanation about the equation for PVL in lines 110-111 in the revised manuscript, as follows. "Finally, the PVL was calculated using the following formula: (number of BLV LTR copies/number of BoLA-DRA copies) × 105 cells."

Line 121: P<0.0?

Answer: In response to the reviewer's comment, we have changed to "P<0.05". in line 120 in the revised manuscript.

Line 127: Delete ", which was taken"

Answer: In response to the reviewer's comment, we have deleted ", which was taken" in line 127 in the original manuscript.

Line 128-129: Delete "an enzyme-linked immunosorbent assay". ELISA appeared earlier already.

Answer: In response to the reviewer's comment, we have deleted "an enzyme-linked immunosorbent assay" in lines 128-129 in the original manuscript.

Line 130: "age of the animal" -> EC-leukosis key?

Answer: Yes, this is the EC-leukosis key, as indicated in your comment and reference number 30.

Line 135: cattle # -> animal #. "cattle" is generally used as a group.

Answer: In response to the reviewer's comment, we have changed to use of animals in lines 134, 136, 138, 139, 140, 141, 155, 156, 162, 167, 181, 182, 208. 210, and 247-249 in the revised manuscript.

Line 154: The closed circle is too small. It should be the same size as the open circle.

Answer: In response to the reviewer's comment, we have changed the size of the closed circle in the Figure legend of Fig.1 in the revised manuscript.

Line 154: Fig 1. Experimental cattle -> experimentally infected cattle. Avoid "experimental cattle". It appears in other places (line136 and so on).

Answer: In response to the reviewer's comment, we have changed to "experimentally infected cattle" in lines 134, 143, 155, 159, 161, 173, 176, and 188 in the revised manuscript.

Line 162: "all the organs, such as" -> "all examined organs including"?

Answer: In response to the reviewer's comment, we have changed from "all the organs, such as" to "all examined organs, including" in lines 163-164 in the revised manuscript.

Line 166: "However", and "particularly"-> these wordings are not appropriate.

Answer: In response to the reviewer's comment, we have deleted "However", and "particularly"- in line 166 in the original manuscript.

Line 204: "This result is supported by" others. This could mean this study was a simple confirmation of previous studies. Novelty in this study should be therefore stated. Also, "Sites of in vivo replication of bovine leukemia virus in experimentally infected cattle" by M. J. Van Der Maaten, J. M. Miller 1978.  should be cited.

Answer: We agree with the reviewer's comment that it should be cited paper "Sites of in vivo replication of bovine leukemia virus in experimentally infected cattle" by M. J. Van Der Maaten, J. M. Miller 1978. In response to the reviewer's comment, we have cited this in References as reference number 32 and changed the order of references in the revised manuscript. Concomitantly, we have added an explanation "This finding correlates well with results reported previously that the initial site of BLV replication was in the spleen rather than the regional lymph node nearest to the inoculation site [32]." In lines 232-234 in the revised manuscript.

Line 206: The "correlation" should be analyzed using such as Pearson's Coefficient.

Answer: We agree with the reviewer's comment that the "correlation" should be analyzed using such as Pearson's Coefficient. In response to the reviewer's comment, we have analyzed the correlation between blood PVL and PVL in all examined organs, including the heart, lung, liver, kidney, abomasum, thymus, spleen, and lymph nodes using such as Pearson's Coefficient and then obtained a strong positive correlation. Therefore, we have added new data as a new Figure 2 and its Figure legend (lines 201-203) in the revised manuscript. In addition, we have added several explanations in the "Result section" (lines 191-198) and "Materials and Methods section" (lines 120-121) in the revised manuscript.

Line 209-211: The sentence is confusing and needs revision.

Answer: In response to the reviewer's comment, we have modified the sentence as follows: "in cattle with a low blood PVL, proviral DNA was only detected in the spleen and lymph nodes, and the blood PVL level was comparable to that of these organs." in lines 225-227 in the revised manuscript.

Line 214-216: The sentence is confusing and needs revision.

Answer: In response to the reviewer's comment, we have modified the sentence as follows: "However, in our study, we detected BLV proviral DNA in the spleen of all animals, and the PVL in the spleen was generally higher than that in the other organs." in lines 230-231 in the revised manuscript.

Line 227: "For example"?

Answer: In response to the reviewer's comment, we have deleted "For example" in line 227 in the original manuscript.

Line 227, 229: Delete "peripheral"

Answer: In response to the reviewer's comment, we have deleted "peripheral" in lines 227 and 229 in the original manuscript.

Line 230: There is a double space between blood and PVL.

Answer: In response to the reviewer's comment, we have excluded space in line 230 in the revised manuscript.

Line 232: "Similarly"?

Answer: In response to the reviewer's comment, we have an exchange from "Similarly" to "In addition" in line 250 of the revised manuscript.

Line 233: "matched blood samples"?

Answer: We apologize for our description making the reviewer confusing. The meaning "matched blood samples" is a blood sample taken from an individual in which milk, nasal mucus, and saliva samples were taken.

Line 232-235: Organize the sentence according to what should be emphasized. It could be divided into two sentences.

Answer: We apologize about our description made the reviewer confusing, we have modified the sentence to include "matched blood samples", as follows. "the BLV provirus can be detected in body fluids such as milk, nasal mucus, and saliva samples of dairy cattle, with PVLs of more than 10,000, 14,000, and 18,000 copies/105 cells in blood samples, respectively [14,15]" in lines 251-253 in the revised manuscript.

Line 243: BLV-sensitive -> BLV-susceptible

Answer: In response to the reviewer's comment, we have an exchange from "BLV-sensitive" to "BLV-susceptible" line 262 in the revised manuscript.

Line 247: "In addition"?

Answer: In response to the reviewer's comment, we have replaced "In addition" with "On the other hand" in line 266 in the original manuscript.

Line 248: "BoLA-DRB3 allele (BoLA-DRB3*1201) -> BoLA-DRB3*012:01

Answer: In response to the reviewer's comment, we have changed to "BoLA-DRB3*012:01", "BoLA-DRB3*016:01", "BoLA-DRB3*005:03", "BoLA-DRB3*015:01" and "BoLA-DQA1*002:04" in lines 262-271 in the revised manuscript.

Line 239-261: This paragraph contains information on DRB3 alleles for cattle used in this study. The information should be included in Table 1 and the method for typing should be described in Materials and Methods. Also, consider changing the names of alleles according to the latest nomenclature site. For example, DRB3*1601 -> DRB3*016:01.

Answer: We deeply apologize for not including information on BoLA-DRB3 alleles for cattle used in this study in Table 1 because we also believe that the BoLA-DQA1 allele is very important for understanding the host factors that are associated with the BLV PVL, and we are planning to type the BoLA-DQA1 allele of the animals used in this experiment in the future.

Line 250: is related -> is associated

Answer: In response to the reviewer's comment, we have replaced "is related" with "is associated" in line 276 in the revised manuscript.

Line 264: "strongly eliminated"?

Answer: In response to the reviewer's comment, we have replaced "strongly eliminated" with "strongly eliminated from the blood by the immune response" in line 283 in the revised manuscript.

Line 276: "specific organs, including the spleen and lymph nodes". What are the specific organs?

Answer: In response to the reviewer's comment, we have replaced "the specific organs" with "particular organs" in line 284 in the revised manuscript.

Line 277. "the level of the virus in the organs was the same as that in the blood". Was it the same? comparable? similar?

Answer: In response to the reviewer's comment, we have changed from "same" to "comparable" in line 296 in the revised manuscript.

Reviewer 2 Report

BLV is serious problem in cattle farm around the world, and It is important to clarify the virus dynamics in the body.

In this study, the authors inoculated individuals with BLV-infected leukocytes to confirm the relationships between the level and distribution of the proviral load after the infection. However, only one dose, 8x107 cells including 4x107 copies of the BLV, was evaluated. This value was so high refered to the introduction, and multiple doses should be evaluated when you mentioned about "correlation". So, I suggest all parts regarding correlations should be changed or multiple dose experiments should be added.

In Materials and Methods section, the authors should describe how to determine the cell numbers in bloods and organs samples. Especially in the organ samples, it should be describe whether cell numbers indicate white bllod cell numbers or organ cell number. When it indecate blood cell number, it also should describe how to collect blood cells from each organs. 

How did you distinguish between remaining donor cells and infection of recipient cells? Since donor cells are injected in very large numbers, antibodies might be induced by viral particles released from the donor cells, so just confirming antibodies is not enough to say that the patient is infected. it should be better to describe when donor cells was disappeared. The sequencing of d-loop region of mitchondrial DNA might help for identification of the origin of cells.

Author Response

Answer to the comments of reviewer #2:

Thank you very much for the helpful and constructive comments. We have amended the manuscript in accordance with these comments and our point-by-point responses are set out below.

BLV is serious problem in cattle farm around the world, and It is important to clarify the virus dynamics in the body.

In this study, the authors inoculated individuals with BLV-infected leukocytes to confirm the relationships between the level and distribution of the proviral load after the infection. However, only one dose, 8x107 cells including 4x107 copies of the BLV, was evaluated. This value was so high refered to the introduction, and multiple doses should be evaluated when you mentioned about "correlation". So, I suggest all parts regarding correlations should be changed or multiple dose experiments should be added.

Answer: We agree with the reviewer’s comment that multiple-dose experiments should be added, but we deeply apologize that we had used only one dose experiment which is 8x107 cells including 4 x 107 copies of the BLV in this study. Instead of multiple-dose experiments, we have analyzed the correlation between blood PVL and PVL in all examined organs, including the heart, lung, liver, kidney, abomasum, thymus, spleen, and lymph nodes using such Pearson's Coefficient and then we have obtained a strong positive correlation, as follows: We constructed a scatter graph and performed linear regression analysis using the PVL data in the blood and each organ of the animals and we obtained a strong positive correlation and the Spearman’s rank correlation coefficient (R) ranged from 0.6787 to 0.9639. Our result showed that there was a correlation between the blood PVL and the biodistribution of proviral DNA. Therefore, we hope to keep a "correlation" between the level and distribution of the proviral load after the infection in this study. Thus, we have added new data as a new Figure 2 and its Figure legend (lines 201-203) in the revised manuscript. In addition, we have added several explanations in the “Result section” (lines 191-198) and “Materials and Methods section” (lines 120-121) in the revised manuscript.

In Materials and Methods section, the authors should describe how to determine the cell numbers in bloods and organs samples. Especially in the organ samples, it should be describe whether cell numbers indicate white bllod cell numbers or organ cell number. When it indecate blood cell number, it also should describe how to collect blood cells from each organs. 

Answer: We apologize for our description making the reviewer confusing. We determined the cell numbers in blood using an automated blood cell counter and we had described it in lines 91-93 in the original manuscript. The cell numbers in BLV proviral load in organ samples using BLV-CoCoMo-qPCR-2 indicate not blood cells but organ cells. Because the organ samples had been collected from exsanguinated animals and blood cells had been removed from the organ samples. Therefore, in response to the reviewer’s comment, we have added an explanation that “Blood cells had removed from the organ samples” in lines 95-96 in the revised manuscript to abolish confusing

How did you distinguish between remaining donor cells and infection of recipient cells? Since donor cells are injected in very large numbers, antibodies might be induced by viral particles released from the donor cells, so just confirming antibodies is not enough to say that the patient is infected. it should be better to describe when donor cells was disappeared. The sequencing of d-loop region of mitchondrial DNA might help for identification of the origin of cells.

Answer: The organ samples are infected with recipient blood cells thereby the organ samples do not include remaining donor cells. On the other hand, it is difficult to distinguish between remaining donor cells and infection of recipient cells in peripheral blood. Fortunately, we had previously determined alleles of bovine leukocyte antigen (BoLA)-DRB3 is a highly polymorphic gene of blood cells from a donor cow persistently infected with BLV using the PCR-sequenced-based typing (SBT) method. Interestingly, we could not detect alleles matching the BoLA-DRB3 alleles of a donor cow in the same DNA when analyzing PVL of all of the 7 experimentally infected cattle used in this study. In all experimentally infected cattle in this study, PVL increased with a much larger amount than inoculated copy numbers from a donor cow. In general, it appears that the virus was eliminated from the blood by the host cell’s immune response after seroconversion. Therefore, infected cells in peripheral blood from experimentally infected cattle may not originate from remaining donor cells.

Round 2

Reviewer 2 Report

In the present infection experiment, the samples were taken within 10 months of infection and the white blood cell counts were in the normal range. It is unclear whether a correlation in the early stages of infection would lead to the development of EBL. In addition, the correlation should be analyzed after logarithmic conversion of the viral load, and it cannot be said that there is a correlation at this analysis. Since these results may mislead the reader, the title should be changed to "Distribution of BLV infection during early stages" etc.

The method used to remove blood cells from the organs should be described in detail. Were the individuals or organs perfused?

Author Response

Answer to the comments of reviewer #2:

Thank you very much for the helpful and constructive comments. We have amended the manuscript in accordance with these comments and our point-by point responses are set out below.

In the present infection experiment, the samples were taken within 10 months of infection and the white blood cell counts were in the normal range. It is unclear whether a correlation in the early stages of infection would lead to the development of EBL. In addition, the correlation should be analyzed after logarithmic conversion of the viral load, and it cannot be said that there is a correlation at this analysis. Since these results may mislead the reader, the title should be changed to "Distribution of BLV infection during early stages" etc.

Answer: We agree with the reviewer comment that it is unclear whether a correlation in the early stages of infection would lead to the development of EBL, and the title should be changed to "Distribution of BLV infection during early stages". Therefore, we have changed the title from “Correlation between the Biodistribution of Bovine Leukemia Virus in the Organs and the Proviral Load in the Peripheral Blood in Experimentally Infected Cattle“ to “Correlation between the Biodistribution of Bovine Leukemia Virus in the Organs and the Proviral Load in the Peripheral Blood during Early Stages of Experimentally Infected Cattle”, and we have also revised to “of BLV infection during early stages“ in other three points in lines 30, 288 and 302-303 in the 2nd revised manuscript.